# Fast Identification Method of Mine Water Source Based on Laser-Induced Fluorescence Technology and Optimized LSTM

Pengcheng Yan [1,2], Xiaofei Zhang [2,*], Xuyue Kan [2], Heng Zhang [2], Runsheng Qi [3] and Qingyun Huang [3]

1 State Key Laboratory of Deep Coal Mining Response and Disaster Prevention and Control, Anhui University of Science & Technology, Huainan 232000, China
2 School of Electrical and Information Engineering, Anhui University of Science & Technology, Huainan 232000, China
3 School of Artificial Intelligence, Anhui University of Science & Technology, Huainan 232000, China
* Correspondence: zxf19971012@163.com; Tel.: +86-177-7521-7001

**Abstract:** There is a great threat to the production safety of coal mines caused by mine water disasters. Traditional identification methods are not adapted to the efficiency of today's coal mining and do not offer the advantage of accurate detection in real-time. In this study, the Mayfly Algorithm (MA) was used to optimize the Long Short-Term Memory (LSTM) network, combined with laser-induced fluorescence technology, to apply it to the identification of mine water sources for the prevention of mine water disasters and post-disaster relief work. Taking sandstone water and goaf water as the original samples, five mixed water samples were also prepared by mixing the sandstone water and goaf water in different proportions, giving a total of seven water samples to be tested. Laser-induced fluorescence technology was used to obtain the fluorescence spectral data of water samples, and then the Linear Discriminant Analysis (LDA) dimensionality reduction algorithm and the Principal Component Analysis (PCA) dimensionality reduction algorithm were used to reduce the dimensions of the original spectral data. Then, three architectures, including LSTM, GA-LSTM (optimization of the LSTM by genetic algorithm) and MA-LSTM were designed to identify mine water sources. Finally, from the results' analysis, MA-LSTM performs best in many aspects after PCA dimensionality reduction and has the best identification effect. These results supported the feasibility of the novel method.

**Keywords:** laser-induced fluorescence; mine water source; LSTM





## 1. Introduction

As one of the major disasters affecting the safety of coal mine production, mine water disasters cause hundreds of millions of economic and property losses every year and pose a great threat to the lives and property safety of mine workers [1–3]. Water disasters in coal mines mostly occur at the coal mining tail entry. When the coal mining channel encounters underground dark rivers, waterlogged caves and water-rich aquifers, there is a massive outflow of water into the coal mining tunnel. In the event of a mine flood, the mine may only be flooded, or the mine may be destroyed and personnel killed. Therefore, prediction and prevention of mine water disasters are very important in coal mining, and the rapid identification of water sources is a powerful tool for solving mine water damage. Reducing the occurrence of mine water disasters is not only necessary for normal operation and production safety, but is also necessary for the safety of mine workers and their property. The main measure used to avoid mine water disasters is to determine the category of the mine water source, to accurately judge the cause of mine water inrush, analyze mine water inrush, find the water source, formulate the mining plan, and design the risk avoidance plan in advance. Therefore, whether the mine water source can be accurately and quickly identified is related to the prevention of water disasters and the implementation and deployment of post-disaster rescue [4–6].

At present, the identification of mine water sources is mainly based on the different concentrations of seven typical representative ions in the water. Other methods include the isotopes, radioactive elements, trace elements and geostatistical simulation [7–13]. However, the traditional identification methods have problems of long detection time and poor flexibility, and cannot accurately complete the mine water-source identification in real-time [14].

In recent years, with the development of laser-induced fluorescence technology, it has been used in many fields. By observing the distribution of copper metal vapor after the arc was extinguished at different times by planar laser-induced fluorescence, Shaogui Ai, Yiping Fan and Yucheng Li found that the copper vapor density reached its peak when the arc extinguished at 7 ms [15]. Minggang Wan, Mingbo Sun and Ge Wu used laser-induced fluorescence to complete the quantitative feature extraction of turbulent mixed flames and improved the visualization of the flame structure [16]. Taylor AT's team analyzed the future applications of laser-induced fluorescence in biochemistry and its development [17]. However, research on laser-induced fluorescence in predicting mine water sources and coal production safety is still relatively scarce and needs improvement. Based on the fluorescence spectral data of different mine water-sources, scientists are still required for the construction of a water-source detection database to prevent water disasters in mines and ensure coal production safety.

The LSTM neural network, a variant of recurrent neural networks (RNNs), makes up for the shortcomings of RNNs in problems such as gradient disappearance and gradient explosion arising from the training process of long sequences [18,19]. The LSTM has been widely used in various fields and has become one of the main algorithms for deep learning [19–22]. The LSTM neural network has long been used in the field of event prediction; however, fluorescence spectral data have more features and the LSTM neural network has problems such as insufficient processing of long sequences, and the model needs to be optimized to improve its prediction accuracy and efficiency [23,24].

The Mayfly Algorithm (MA) is a new intelligent optimization algorithm proposed in 2020, which simulates the social behavior of mayflies and combines the main advantages of group intelligence algorithms and evolutionary algorithms, with good performance in terms of convergence speed and optimization-seeking accuracy. It abstractly models the actions of mayflies during the solution process, enhancing the balance between the exploratory and exploitative aspects of the algorithm and avoiding the algorithm falling into a local optimum [25–27]. Rajakumar MP and Sonia R et al. used the MA to optimize the dual deep-learning function to improve the accuracy of chest X-ray detection of tuberculosis [28]. Yuhu Liu and Yi Chai's team used an improved MA combined with energy spectrum statistics to diagnose bearing faults more accurately [29].

According to the above problems, this paper proposes to use laser-induced fluorescence technology to obtain fluorescence spectral data of different mine water-sources, reduce the dimensionality of the spectral data, train and predict through MA-optimized LSTM to finally determine the type of water sources and ultimately to improve the accuracy and overall efficiency of mine water-source identification.

## 2. Materials and Methods

### 2.1. Sample Preparation

The experimental water samples consisted of seven categories, including goaf water and sandstone water, and another five categories were water samples that mixed goaf water and sandstone water in different ratios (10:4, 10:7, 10:10, 7:10, 4:10), each category containing 30 water samples each. The water samples used for the experiments were collected from the Huainan mine. After the water sample types were configured, the prepared water samples were prepared and labelled (1, 2, 3, 4, 5, 6, 7) according to the different mixing ratios. Finally, the experimental water samples were sealed and stored in glass bottles, protected from light.

The specific wavelength laser worked normally with the support of a regulated power supply, generating a specific wavelength laser, which was transmitted via an optical fiber with corresponding characteristics to a miniature fluorescence probe, which injected the laser into the measured water body, and then received the fluorescence generated by the excited radiation of the measured water body. After some filtering and use of filtering modules to obtain the required spectral band, it was then transmitted to the spectrometer via the corresponding optical fiber for photoelectric signal conversion and analogue-to-digital conversion, etc., which finally provided the spectral data of the water sample for subsequent analysis.

### 2.2. Apparatus

The instruments used for the experiments included a 405 nm single-mode laser from Hangzhou SPL Photonics Co., Ltd. (Hangzhou, China), with a laser power setting of 100 mW, an immersion fluorescence probe (FPB-405-V3) from Guangzhou SPL Photonics, and a miniature fiber optic spectrometer (USB2000+) from Ocean Optics (Orlando, FL, USA), with a setting of Integration time 1 s, sampling spacing 340–1020 nm, spectral resolution 1 nm. Our measurements were obtained with no expensive equipment.

The analysis after obtaining the spectral data was completed on a computer with a Windows 11 operating system. The hardware on this computer was an Intel (R) Core (TM) i7-8750H CPU@2.20 GHz and 8.00 GB RAM. The code designed for this analysis was written in Python 3.9 and the deep model training was performed using the Python library Keras.

### 2.3. Dimensionality Reduction

The original spectral data had 2048 eigenvalues, and such high-dimensional samples were sparse and it was difficult to find data features. To improve the recognition accuracy of the model, and the generalization ability of the model, this paper used PCA and LDA to reduce the dimensionality of the original spectral data to three-dimensions to compare the effect of dimensionality reduction clustering. After that, the reduced, dimensional fluorescence spectral data were divided into a training set and a prediction set according to the ratio of 2:1.

### 2.4. Deep Learning

The LSTM neural network was trained by mapping the training set to its corresponding labels. In addition to the original LSTM neural network, an LSTM neural network using MA was added to optimize the hyperparameters of the model to enhance the recognition effect and improve the generalization and robustness of the model. To compare the optimization effect, a genetic algorithm (GA) was used to optimize the LSTM to compare the recognition effect and the model's strengths and weaknesses from different evaluation perspectives [30,31].

First, initialize the female mayfly and male mayfly populations and set the optimization boundary of the optimized super parameters, including the learning rate, training times, batch size, the number of nodes in the two LSTM hidden layers, and the number of nodes in the full connection layer. The optimization process is shown as follows:

1. Seven-dimensional vector candidate solution $x = (x_1, \ldots, x_7)$.
2. Mayfly velocity $v = (v_1, \ldots, v_7)$, defined as its position change.
3. Each mayfly automatically adjusts its trajectory to its current best position (*pbest*) and the best position (*gbest*) any mayfly has achieved so far.
4. Calculate fitness values, sort them, and obtain *pbest* and *gbest*.

5. Update the position of male mayflies and female mayflies in turn, and make them mate.

(a) Male mayfly movement: $x_i^t$ is the position of male mayfly $i$ in the search space at the time step $t$, add $v_i^{t+1}$ to the current position to change the position, expressed as:

$$x_i^{t+1} = x_i^t + v_i^{t+1} \tag{1}$$

The velocity of male mayflies is calculated as:

$$v_{ij}^{t+1} = v_{ij}^t + \alpha_1 e^{-\beta r_p^2}\left(pbest_{ij} - x_{ij}^t\right) + \alpha_2 e^{-\beta r_g^2}\left(gbest_{ij} - x_{ij}^t\right) \tag{2}$$

where $v_{ij}^t$ is the velocity of the ephemera at time $t$ in dimensions $i$ and $j$; $x_{ij}^t$ represents the position at time $t$; $\alpha_1$ and $\alpha_2$ is the positive attraction coefficient of social effect, which is 1.0 and 1.5, respectively; $\beta$ is the visibility coefficient of mayfly, which is 2.0, thus controlling the visibility range of mayfly; $r_p$ represents the distance between the current position and $pbest$, $r_g$ represents the distance between the current position and $gbest$. The distance is calculated as:

$$\| x_i - X_i \| = \sqrt{\sum_{j=1}^{n}\left(x_{ij} - X_{ij}\right)^2} \tag{3}$$

The best mayflies in the colony constantly change their velocity, which is calculated as follows:

$$v_{ij}^{t+1} = v_{ij}^t + d * r \tag{4}$$

where $d$ is the dance coefficient, which is 5.0. $r$ is a random number between $-1$ and 1. This up and down movement introduces random elements into the algorithm.

(b)     Female mayfly movement: Female mayflies do not congregate, but fly to males to reproduce. $y_i^t$ is the mayfly $i$ at time $t$. Its position is updated by increasing the speed:

$$y_i^{t+1} = y_i^t + v_i^{t+1} \tag{5}$$

According to their robust attributes, the best females will be attracted by the best males, the second-class females will be attracted by the second-class males, and so on. Therefore, in the minimization problem, the velocity is:

$$v_{ij}^t = \begin{cases} v_{ij}^t + \alpha_3 e^{-\beta r_{mf}^2}\left(x_{ij}^t - y_{ij}^t\right), \; if \; f(y_i) > f(x_i) \\ v_{ij}^t + fl * r, \; if \; f(y_i) \leq f(x_i) \end{cases} \tag{6}$$

where $v_{ij}^t$ stands for velocity and $y_{ij}^t$ represents the position of mayfly. $\alpha_3$ is a positive coefficient, which is 1.5. $\beta$ is a fixed visibility coefficient, which is 2.0. $r_{mf}$ represents the distance between female mayflies and male mayflies, and its calculation method is the same as the distance calculation formula. $fl$ is a random walk coefficient, which is 1.0. $r$ is a random number in the range $[-1,1]$.

(c)     Mayflies mating: Select a parent from the male population and female population based on fitness function, and cross to produce two offspring as follows:

$$offspring1 = L * male + (1 - L) * female \tag{7}$$

$$offspring2 = L * female + (1 - L) * male \tag{8}$$

where $male$ is the male parent; $female$ is the female parent; and $L$ is a random number in a specific range.

6.     Calculate fitness, update $pbest$ and $gbest$.

If the stop conditions are met, exit and output the results. Otherwise, repeat steps (2) to (6).

*2.5. Model Assessment*

The evaluation of the experimental model was used to evaluate and compare the prediction results of the validation set with the real value, the confusion matrix of the

multi-classification model, the changing trend of the accuracy of the training samples, and the changing trend of the loss function. The closer the prediction result is to the real value, the better the model is on the prediction effect. The fewer erroneous judgment that result in the confusion matrix, the better the classification effect of the model. The trend of accuracy is that the fewer iterations the model has, the better the performance of the model will be. The loss function belongs to the sparse categorical cross-entropy loss function, that is, the fewer the number of iterations, the faster the loss function curve tends to converge and stabilize, and the model has better robustness and stability.

## 3. Results

### 3.1. Spectral Data

The spectral characteristics of a substance reflect the properties and state of the substance itself, which vary from substance to substance. The spectral characteristics of a body of water are determined by the absorption and scattering properties of the light radiation by the various optically active substances in it. The laser-induced fluorescence technique is the basis for laser detection of water bodies by measuring the spectral characteristics of a body of water in a certain wavelength range, which is achieved by analyzing the spectral characteristics formed by the absorption and scattering of solar radiation energy by the water body. The fluorescence spectra of the water samples obtained are shown in Figure 1. From Figure 1, it can be seen that in the 420–650 nm band, the difference between the spectral data of the water samples is relatively obvious. The original spectral data are not blurred, there is no excessive redundancy and interweaving, there are large gaps between the fluorescence spectra of water samples numbered 1 and 7 and the rest of the numbered fluorescence spectra, and the fluorescence spectra of water samples numbered 2–6 have a small gap. While in the 340–420 nm and 650–1020 nm bands, the images are very similar. This is because the fluorescence spectrum intensity increases as the proportion of goaf water increases. The sandstone water locks out the strata at a deeper level, resulting in less organic matter and its internal composition is mainly inorganic. In contrast, the goaf water is in an anthropogenic area, and its water contains more organic matter, and its composition is relatively complex, so the fluorescence spectrum features are more obvious. The difference in fluorescence spectra between water samples is due to the difference in the composition and concentration of chemicals in the water, which is the theoretical basis for research into the identification of water sources in coal mines.

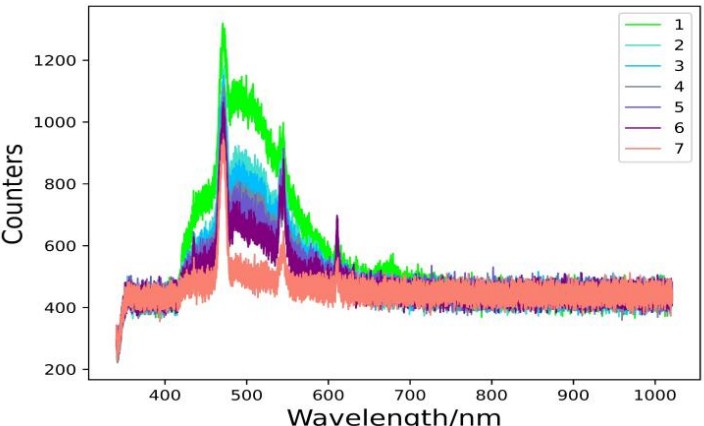

**Figure 1.** Original spectra.

### 3.2. Dimensionality Reduction

Figure 2 shows the clustering effect of LDA and PCA. It can be seen from the figures that the fluorescence spectral data after dimensionality reduction by PCA have an obvious clustering effect, while the spectral data after dimensionality reduction by LDA are more discrete. Figure 3 shows the respective contribution of the three eigenvalues after LDA and

PCA dimensionality reduction. It can be seen that after the dimensionality reduction in LDA and PCA, the first eigenvalue X has the highest contribution and accounts for the highest proportion of information, while the cumulative contribution of LDA is significantly higher than PCA.

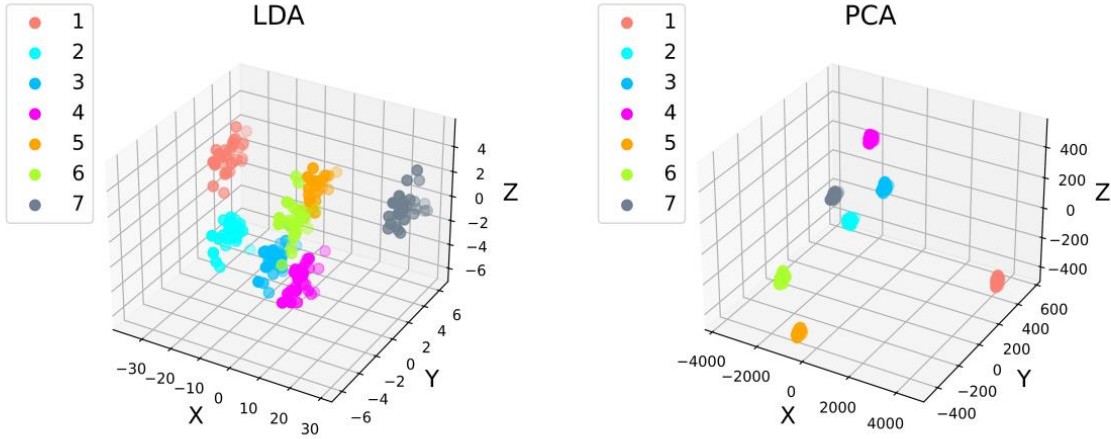

**Figure 2.** LDA and PCA dimensionality reduction.

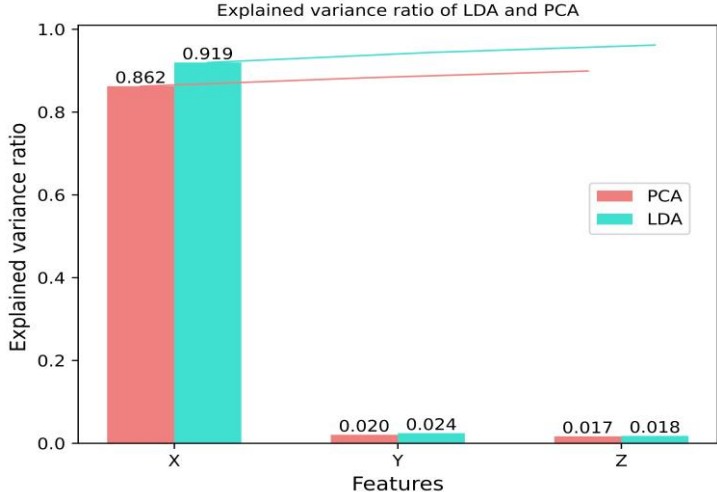

**Figure 3.** Explained variance ratio of LDA and PCA.

### 3.3. Models Performance

From (a) and (b) in Figure 4 for different water-source identification models built after LDA and PCA dimensionality reduction, it can be seen that after LDA dimensionality reduction processing the prediction of the three models is unstable and fluctuates greatly, among which the MA-optimized LSTM model has a more obvious improvement and optimization of the prediction effect compared with the original LSTM. The GA optimization effect is not ideal, and the prediction difference is not significant compared with the original LSTM. In the LSTM model built after PCA dimensionality reduction, the MA-LSTM prediction effect is the best, the closest to the true value, and matches the true value. The GA also has a more obvious improvement in the LSTM model. The original LSTM model has the worst prediction effect and is not well-differentiated in the water sample prediction of water samples numbered 2 and 3.

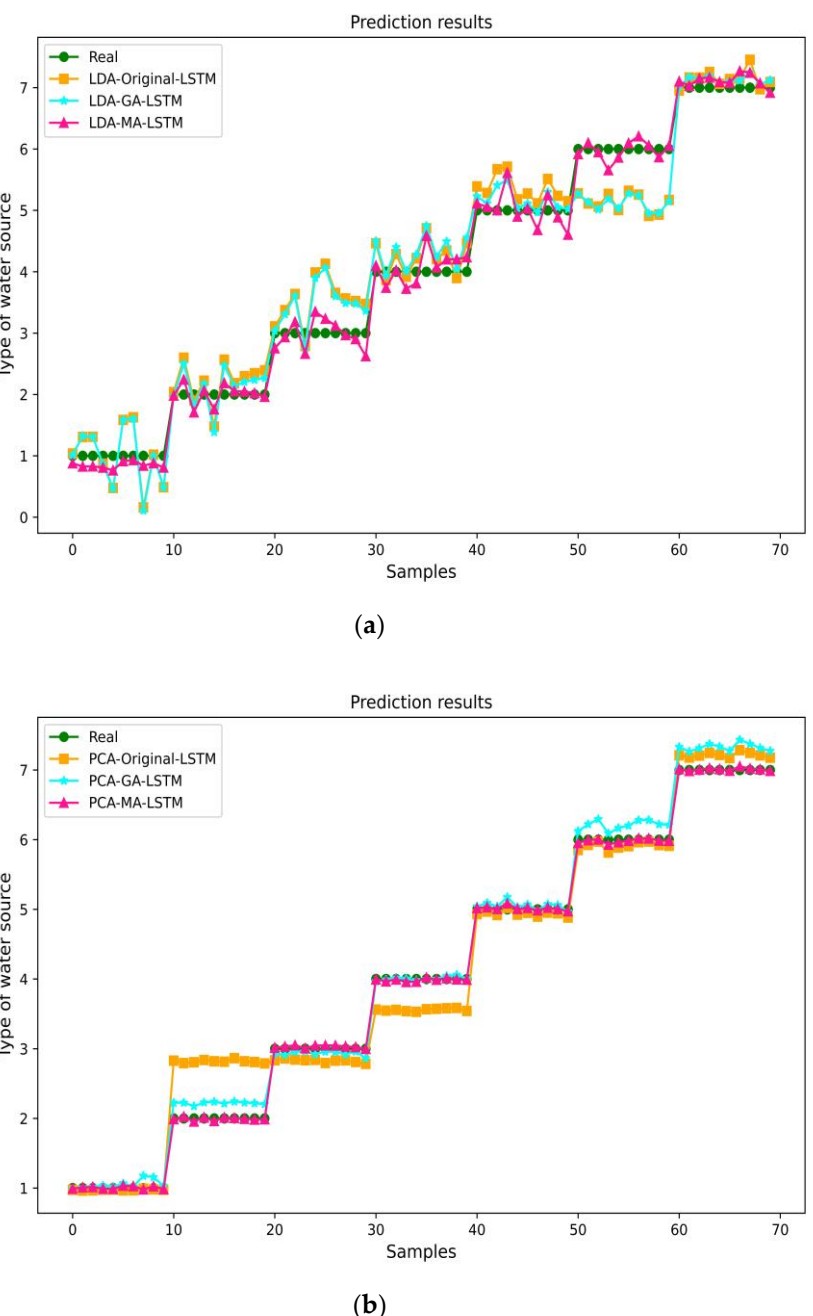

**Figure 4.** Prediction results for the validation set of different models: (**a**) Prediction results of different models after LDA dimensionality reduction; (**b**) Prediction results of different models after PCA dimensionality reduction.

Figure 5 shows the multi-classification confusion matrix of different water-source identification models. It can be seen that the LSTM classification models after PCA and LDA dimensionality reduction all have good classification results and maintain a high recognition accuracy. In the fluorescence spectra of water samples after LDA dimensionality reduction, when using the original LSTM neural network, the water samples labelled as 3 and 6 were misclassified as water samples 4 and 5, respectively, and there was a water sample labelled as 6 that was misclassified as 5 after GA-LSTM training. All the water samples in the test set were identified accurately after MA-LSTM training. In the fluorescence spectra of water samples after PCA dimensionality reduction, the original LSTM, GA-LSTM, and MA-LSTM were all correct for the test set after training.

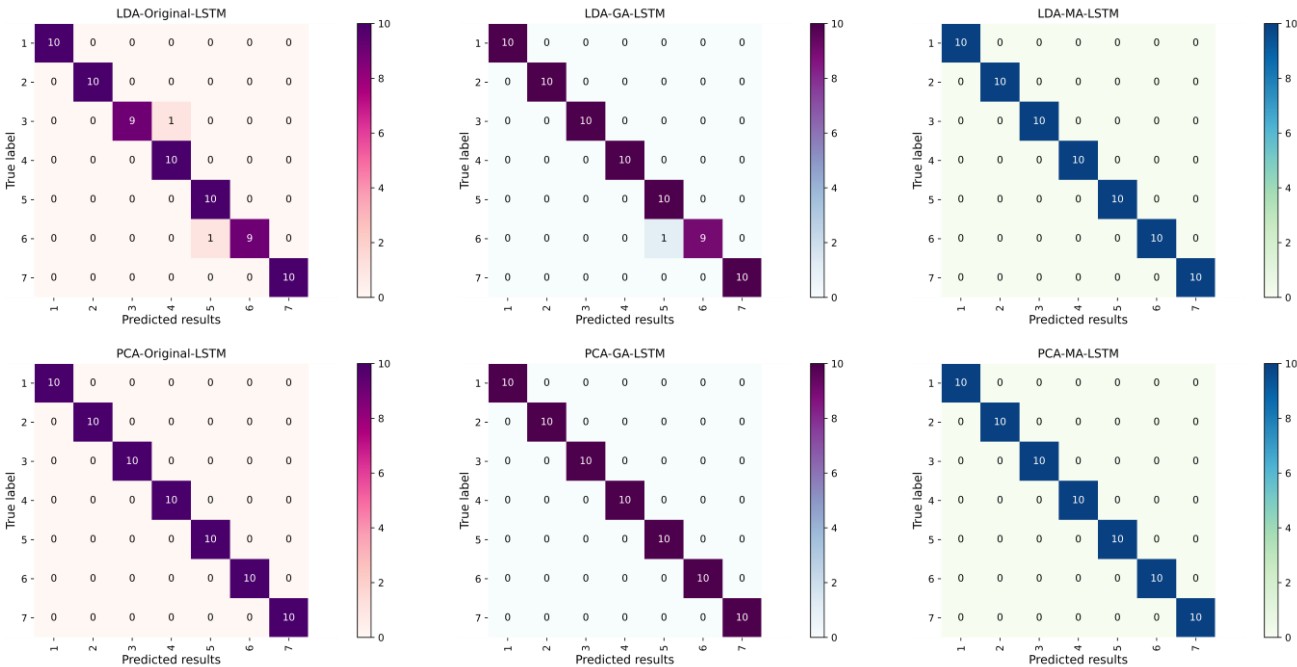

**Figure 5.** Multi-category confusion matrix.

The precision, recall, and F1-score of the macro-average derived from the multiclassification confusion matrix of each model are shown in Table 1, and it can be seen that the precision, recall, and F1-score of the Original-LSTM and the GA-LSTM built by LDA dimensionality reduction are 0.97 and 0.99, respectively. It can be seen that MA optimizes the model better than GA, and PCA is more suitable for processing the fluorescence spectral data of the experimental water samples.

**Table 1.** Macro-average of different LSTM models.

| Identification Models | Precision | Recall | F1-Score | Support |
|---|---|---|---|---|
| LDA-Original-LSTM | 0.97 | 0.97 | 0.97 | 70 |
| LDA-GA-LSTM | 0.99 | 0.99 | 0.99 | 70 |
| LDA-MA-LSTM | 1.00 | 1.00 | 1.00 | 70 |
| PCA-Original-LSTM | 1.00 | 1.00 | 1.00 | 70 |
| PCA-GA-LSTM | 1.00 | 1.00 | 1.00 | 70 |
| PCA-MA-LSTM | 1.00 | 1.00 | 1.00 | 70 |

The trends in the accuracy of the training set and the sparse categorical cross-entropy loss function of the LSTM, GA-LSTM, and MA-LSTM water-source identification models were built after dimensionality reduction by LDA and PCA and are shown in Figures 6 and 7, which show that the LSTM water-source identification model optimization by MA under the same dimensionality reduction method reaches 100% accuracy in the training process the fastest, and the cross-entropy loss function also converges at the fastest convergence. The GA optimization algorithm was not as good as the MA optimization algorithm in terms of optimization effect. In a cross-sectional comparison between the LDA and PCA dimensionality reduction algorithms for fluorescence spectral data, the PCA achieved an accuracy of 1.0 in fewer iterations than the LSTM, GA-LSTM, and MA-LSTM. Neural network models built from the LDA dimensionality reduction processed spectral data during training, and the loss function converged in fewer iterations, with better overall model efficiency and performance.

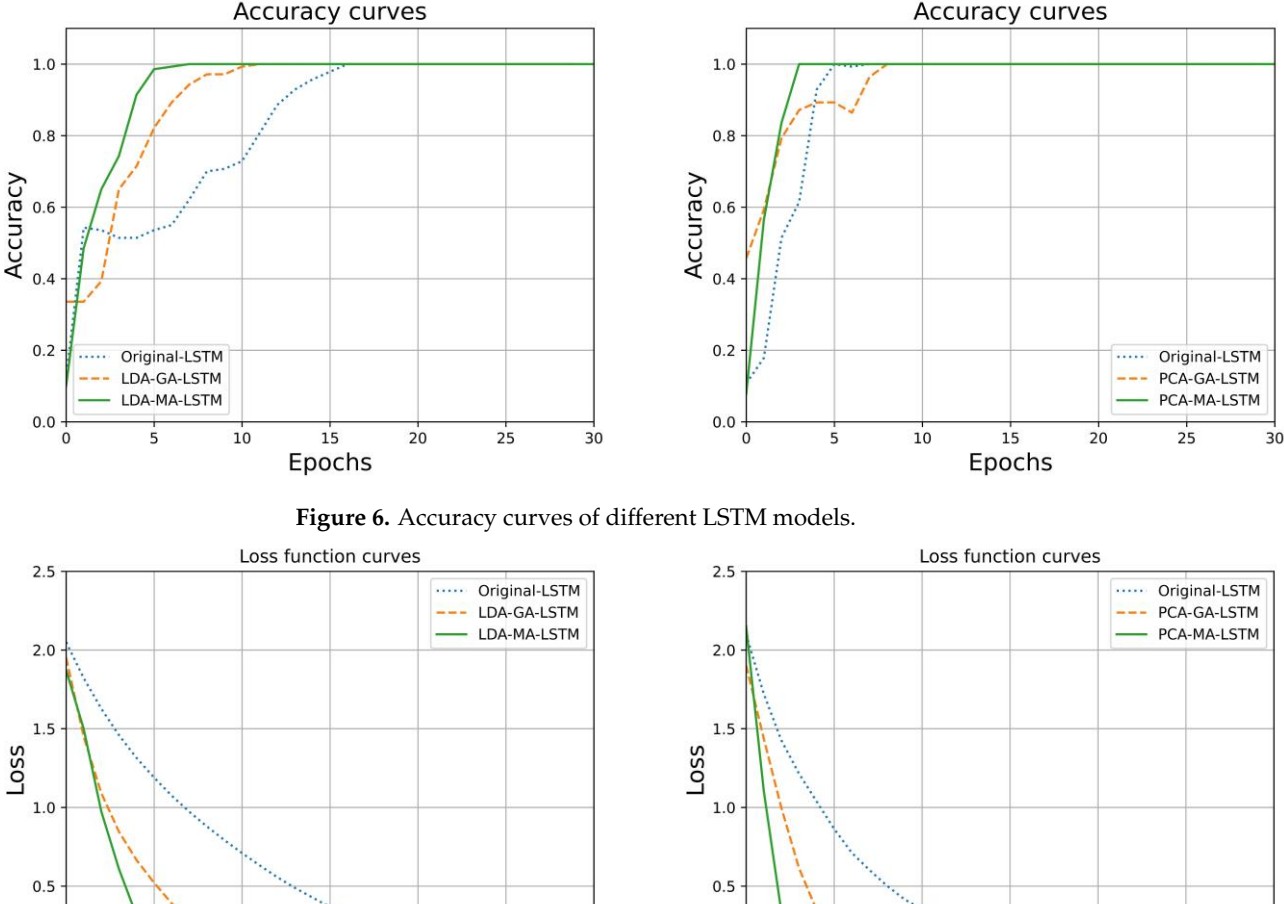

**Figure 6.** Accuracy curves of different LSTM models.

**Figure 7.** Loss curves of different LSTM models.

In the performance comparison of the six water-source identification models, the MA-LSTM mine water-source identification model built after the PCA dimensionality reduction process was the best in terms of water source prediction, identification accuracy, the trend of change in the accuracy of the model training set, and the trend of change in the loss function, and had the most outstanding overall performance. The MA optimization algorithm was compared with the GA optimization algorithm; the MA optimization algorithm had better optimization results than the GA optimization algorithm for the LSTM mine water-source identification model, which was more suitable for improving the performance of the mine water-source identification model.

## 4. Discussion

In this paper, we built various water-source identification models using the different fluorescence spectral characteristics of different mine water-sources and improved the performance of the identification models after model optimization. By comparing the prediction results and the multi-classification confusion matrix of the validation set under different models, we conclude that the MA-LSTM water-source identification model with PCA dimensionality reduction has the best identification performance.

In contrast to the previous research, water source identification often relies on water chemistry analysis, which ensures a high level of accuracy. The application of laser-induced fluorescence technology to water source identification has also been developed in recent years, and by combining it with algorithmic research, good recognition accuracy can also be achieved. In this study, by optimizing the neural network algorithm and combining

it with laser-induced fluorescence technology, we not only achieved fast and accurate water-source identification, but also improved the overall performance of the model, with a small amount of training, to achieve the best performance of the neural network model. The research has, to a certain extent, solved the problem of time-consuming and inefficient water-source identification. Using multidisciplinary intersection, artificial intelligence and laser-induced fluorescence technology are combined to achieve good application in the field of mine water-source identification, improving the safety and efficiency of coal mine production, and safeguarding the lives and property of coal mine workers.

At this stage, the experimental samples were taken from the working area of the authors of this paper, and the experimental samples were small. There were also differences in the types of water sources in different mining areas, and their fluorescence spectral data were also somewhat different. These are the limitations of this experiment. To address the limitations, the water source database can be expanded in the future to ensure the adequacy and diversity of the experimental data.

Most of the work in this study has been completed on the characterization of water source types and future research will be directed towards changing the water-source rationing scheme from qualitative to quantitative analysis of water sources. Dynamic analysis of current trends in water-source fluorescence spectra over time, combined with water source ratios, will establish temporal growth curves for different hazardous water-sources and predict the type and timing of possible water surges.

## 5. Conclusions

The experiments used sandstone water and old hollow water from Huainan mines as the original samples and mixed the two types of water in different proportions to create five additional water samples, resulting in a total of seven water samples, and used the laser-induced fluorescence technique to identify and analyze the fluorescence spectra of the seven water samples. The obtained fluorescence spectra of each water sample were subjected to LDA in PCA dimensionality reduction, and the reduced fluorescence spectra data were divided into the training set and validation set according to a fixed ratio, and the LSTM neural network was used. LSTM, GA-LSTM, and MA-LSTM water identification models were built using the GA optimization algorithm and the MA optimization algorithm, respectively. Finally, the optimal water-source recognition model for the mine was selected by comparing four aspects: the prediction of the validation set; the classification result of the validator; the changing trend of the accuracy of the training set; and the changing trend of the loss function. The following conclusions can be drawn from the experimental results:

(1) The MA-LSTM mine water-source identification model built from the fluorescence spectra of water samples after PCA dimensionality reduction processing has the closest prediction to the actual value, the best identification effect, the highest training efficiency, and the best performance.

(2) Under the LSTM, GA-LSTM, and MA-LSTM models, the model built from the fluorescence spectra after dimensionality reduction by PCA performed better than that after dimensionality reduction by LDA, and PCA was more suitable for dimensionality reduction in the fluorescence spectra of mine water sources than LDA, which improved the overall performance of the water-source identification model.

(3) Among the PCA and LDA dimensionality reduction algorithms, the MA optimization LSTM water-source recognition model is better than GA, and the MA optimization algorithm is more suitable for optimizing the LSTM water-source recognition model and improve the generalization ability and recognition efficiency of the LSTM water-source recognition. It is verified that the MA-LSTM recognition model after PCA dimensionality reduction has the best recognition effect, the best performance, and has certain reliability for mine water-source recognition.

**Author Contributions:** P.Y.: investigation, funding acquisition; X.Z.: funding acquisition, supervision, writing–review and editing; H.Z.: data curation; X.K.: software, methodology; R.Q.: validation; Q.H.: conceptualization. All authors have read and agreed to the published version of the manuscript.

**Funding:** Funding Project for Postdoctoral Research in Anhui Province (2019B350), Key Project of National Key R&D Program (2018YFC0604503), Youth Project of Natural Science Foundation of Anhui Province (1808085QE157).

**Data Availability Statement:** The data used in this paper are not a public dataset and are not suitable for publication. The data used in this paper were obtained by the authors involved in this paper by taking them in an experimental setting and are the property of themselves and their team.

**Acknowledgments:** We thank Huainan Mining District for providing the experimental samples and Anhui University of Science and Technology College of Electrical and Information Engineering for providing the experimental equipment.

**Conflicts of Interest:** The authors declare no conflict of interest.

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
