# Peer review of "Fast Identification Method of Mine Water Source Based on Laser-Induced Fluorescence Technology and Optimized LSTM"

_water, doi:10.3390/w15040701_

Round 1
Reviewer 1 Report
Reviewer´s comment on the paper:
“Fast identification method of mine water source based on laser-induced fluorescence technology and optimized LSTM”
Submitted by Pengcheng Yan et al to Water.
The manuscript presents a case study using the Mayfly Algorithm (MA) to optimize the Long Short-Term Memory (LSTM) network, combined with laser-induced fluorescence technology to identify mine water source in Huainan mine (China). I consider the work fits into the scope and could be interesting for readers of Water. The manuscript shows a great amount of effort; however, I have some comments which are listed below:
(i) Change golf by goaf in the 6th line of the Abstract. Besides, laser should be Laser in the following line. The manuscript is full of typo mistakes.
(ii) What is the novelty of this work? It should be clearly state in the abstract, introduction and conclusions. Morevover, what are the main implications of this work? Finally, is this work aimed to present a new methodology or is a case study? This points are extremely important in my humble opinion.
(iii) Other method to identify and model mine water is geostatistical simulation. Please, consider it and the following references:
Barral et al (2021). Hydrochemical evolution of the Reocín mine filling water (Spain). Environmental Geochemistry and Health, 43(12), 5119-5134.
Barral et al (2021). Spatio-temporal geostatistical modelling of sulphate concentration in the area of the Reocín Mine (Spain) as an indicator of water quality. Environmental Science and Pollution Research, 29(57), 86077-86091.
(iv) The format to write references 13, 14, 26 y 27 is wrong.
(v) State-of-arte is poor in my opinion. The employed techniques are well-explained, however, the introduction of the problem, mine water, lacks of theoretical basis with suitable references.
(vi) Section 2.1 needs more information. Information of the mine and the sampling procedure.
(vii) Section 2.3 and 2.4 need references. Have these techniques been applied to mine water before?
(viii) I think Results section is well-written and presented, however, I cannot find a real discussion section supported by the state-of-art. This section must be-rewritten.
(ix) Conclusions must be changed considering comments (ii), (v) and (viii).
Final recommendation
Based on the previous comments, I recommend that the Authors revise the manuscript improving all the points questioned above and resubmit it for another round of review. My decision is “Major revision”.
Author Response
Response to Reviewer 1 Comments
Point 1: Change golf by goaf in the 6th line of the Abstract. Besides, laser should be Laser in the following line. The manuscript is full of typo mistakes.
Response 1: Thanks for the suggestion, changes have been made based on your comments.
Point 2: What is the novelty of this work? It should be clearly state in the abstract, introduction and conclusions. Morevover, what are the main implications of this work? Finally, is this work aimed to present a new methodology or is a case study? This points are extremely important in my humble opinion.
Response 2: The novelty of this study lies in the application of Laser-induced fluorescence technology in mine water hazard prevention and control, and the rapid adjustment of coal opening and testing strategies through real-time and accurate identification of mine water sources. As a major dangerous accident accompanied by the process of coal mining, mine water damage threatens the life and property safety of production workers. The main significance of this work is to realize the rapid identification of mine water sources in real time, so as to do a good job in the prevention and control of water hazards in the process of mine mining in advance, reduce the risk of mine water damage, and also the need for post-disaster rescue work after water damage. The main purpose of this work is to propose more optimal solutions to the original problems.
Point 3: Other method to identify and model mine water is geostatistical simulation. Please, consider it and the following references:
Barral et al (2021). Hydrochemical evolution of the Reocín mine filling water (Spain). Environmental Geochemistry and Health, 43(12), 5119-5134.
Barral et al (2021). Spatio-temporal geostatistical modelling of sulphate concentration in the area of the Reocín Mine (Spain) as an indicator of water quality. Environmental Science and Pollution Research, 29(57), 86077-86091.
Response 3: Thank you for your comments, have referred to your references.
Point 4: The format to write references 13, 14, 26 y 27 is wrong.
Response 4: Thanks for the suggestion, changes have been made.
Point 5: State-of-arte is poor in my opinion. The employed techniques are well-explained, however, the introduction of the problem, mine water, lacks of theoretical basis with suitable references.
Response 5: Thank you for your comments. The author makes appropriate additions to the theoretical basis of mine water.
Point 6: Section 2.1 needs more information. Information of the mine and the sampling procedure.
Response 6: Thanks for the suggestion, which the author has supplemented in the corresponding section.
Point 7: Section 2.3 and 2.4 need references. Have these techniques been applied to mine water before?
Response 7: In section 2.3 and 2.4, the author also carried out some research, mainly in part 2.4, the author optimized the algorithm visibly to improve the performance of the water source identification model.
Point 8: I think Results section is well-written and presented, however, I cannot find a real discussion section supported by the state-of-art. This section must be-rewritten.
Response 8: Thanks for the suggestion, the author made the corresponding corrections in the discussion section.
Point 9: Conclusions must be changed considering comments (ii), (v) and (viii).
Response 9: Thank you for your advice and guidance.

Reviewer 2 Report
The manuscript presents a fluorescence spectra analysis study based on machine learning. It could be of interest to many readers. It consists of two logical parts:
1. Fluorescence measurements, which are very naive; almost on the verge of being totally incorrect.
2. Machine learning analysis, which is very interesting; and very informative for readers. However, my personal opinion is that authors use “heavy artillery to kill a fly”. Nevertheless, the paper should be published after revision.
I have tried to see from which scientific field are authors, to see that after their names (after paper title) there are letters, however affiliations are denoted by numbers, so I guess that “a” corresponds to “1”, etc….. Only one author is somehow connected with fluorescence, I suppose…. The author contributions statement at the end of text confirms this.
My comments in regard the fluorescence measurements:
1. Caption of Fig 1 should be “Original spectra”. Very indicative mistake.
2. Spectra in Fig 1 have very large background, which is not treated. It is obvious that the fluorescence signals were very small, so the unusually very long integration time of 1 s was used, resulting in a very strong background signal. The fluorescence should exist only above 405 nm, bellow 405 nm it is just a noise. It could be seen that the very strong noise is spread equally on the whole measured spectrum.
3. The wide emission band starting at 405 nm, taking almost all visible part of the signal is probably some kind of inelastic scattering of photons from solid particles in water. My guess is that the signals differ in regard concentration of solid particles in water (optical transparency of measured samples). It is true that authors mention chemical composition and concentration. If authors could use the time-resolved fluorescence measurements, they probably will see that the wide emission band lasts in time only as the laser excitation lasts, implying that it is not fluorescence, or that the fluorescence is very short-lived, which could correspond to organic materials.
4. There are three peaks on all spectra. The peaks were not commented or identified by authors. I suppose that these peaks actually contribute to machine learning analysis, because PCA removes the mean and seeks for variations of data. However, I suppose that even ordinary peak intensity ratio analysis will show the difference between measured samples.
5. I guess that Oceanic, USA means Ocean Optics, USA?
6. I could not find data for Guangzhou SPL Photonics FPB-405-V3 immersion probe. I suppose the probe, connected to the Ocean Optics spectrometer, was immersed in water sample,? So the water sample was not placed in cuvette, actually the probe was immersed in the glass bottle? Attenuating the laser excitation by scattering?
My comments in regard the machine learning:
1. PCA nicely grouped, or clustered (I am not sure is the “clustering effects” correctly used by authors) measured results, so what is the point of further killing a fly with heavy machine learning techniques? I see it as only a well written education of readers.
2. In my experience on using ML for analysis of fluorescence spectra, the simple clustering algorithm will provide the same results.
3. I am not sure is it correct, probably just misleading to say “The original spectral data had 2048 EIGENVALUES”. Anyway, useful part of the spectrum is limited to much less than 2048 points, between 405 nm and less that 700 (650) nm; the spectra should be simply truncated before further analysis.
My final comment is that the strong point of this paper is that is proves that it is possible to obtain correct conclusions from poorly measured results using artificial intelligence.
Author Response
Response to Reviewer 2 Comments
Point 1: Fluorescence measurements, which are very naive; almost on the verge of being totally incorrect.
Response 1: Thank you for your comments. Regarding point 1, the authors believe that the identification of water sources by laser-induced fluorescence technology is supported by theory and is not a wrong method or viewpoint. Laser-induced fluorescence technology has been used earlier in the academic community to identify edible oil adulteration and combustion diagnosis. Therefore, the use of laser-induced fluorescence technology for water source identification in mine bursts has scientific basis and theoretical support.
Point 2: Machine learning analysis, which is very interesting; and very informative for readers. However, my personal opinion is that authors use “heavy artillery to kill a fly”. Nevertheless, the paper should be published after revision.
Response 2: Thank you for your comments, for point 2, the author's research on this topic is to fully realize industrial automation, and the use of optimized LSTM neural networks is to improve the overall recognition performance of the model and prepare for the next step.
Reviewer 2 comments in regard the fluorescence measurements:
Point 1: Caption of Fig 1 should be “Original spectra”. Very indicative mistake.
Response 1: The author has made the correction, thank you for correcting it.
Point 2: Spectra in Fig 1 have very large background, which is not treated. It is obvious that the fluorescence signals were very small, so the unusually very long integration time of 1 s was used, resulting in a very strong background signal. The fluorescence should exist only above 405 nm, bellow 405 nm it is just a noise. It could be seen that the very strong noise is spread equally on the whole measured spectrum.
Response 2: Thank you for your comment, the author mainly applies laser-induced fluorescence technology in the field of mine water hazard control, which mainly uses different water sources to present different spectra of this characteristic. The integration time of 1s is set according to the actual situation of the experiment, and the corresponding adjustment is made according to the situation of the actor.
Point 3: The wide emission band starting at 405 nm, taking almost all visible part of the signal is probably some kind of inelastic scattering of photons from solid particles in water. My guess is that the signals differ in regard concentration of solid particles in water (optical transparency of measured samples). It is true that authors mention chemical composition and concentration. If authors could use the time-resolved fluorescence measurements, they probably will see that the wide emission band lasts in time only as the laser excitation lasts, implying that it is not fluorescence, or that the fluorescence is very short-lived, which could correspond to organic materials.
Response 3: The authors believe that the differences in fluorescence spectra are mainly due to differences in the concentration of organic and fluorescent substances in different water sources and mixed water samples. The time-resolved fluorescence measurement method you mentioned, the author will consider using it at its discretion, thank you for your suggestion.
Point 4: There are three peaks on all spectra. The peaks were not commented or identified by authors. I suppose that these peaks actually contribute to machine learning analysis, because PCA removes the mean and seeks for variations of data. However, I suppose that even ordinary peak intensity ratio analysis will show the difference between measured samples.
Response 4: Thanks for the suggestion, after looking at the values of these three peaks, the difference in some values is not significant, especially the spectral peak on the far right. The author believes that the differences in the bands mentioned in the paper are more obvious, and combined with the consideration and analysis of all data, it is more convincing and universal.
Point 5: I guess that Oceanic, USA means Ocean Optics, USA?
Response 5: Thanks for the suggestion, the author has made changes in the article.
Point 6: I could not find data for Guangzhou SPL Photonics FPB-405-V3 immersion probe. I suppose the probe, connected to the Ocean Optics spectrometer, was immersed in water sample,? So the water sample was not placed in cuvette, actually the probe was immersed in the glass bottle? Attenuating the laser excitation by scattering?
Response 6: Thank you for your comments. The fluorescence probe used in the experiment is Hangzhou SPL Photonics FPB-405-V3, not Guangzhou. Experimental methods have mentioned and introduced in the article.
Reviewer 2 comments in regard the machine learning:
Point 1: PCA nicely grouped, or clustered (I am not sure is the “clustering effects” correctly used by authors) measured results, so what is the point of further killing a fly with heavy machine learning techniques? I see it as only a well written education of readers.
Response 1: Thanks for your suggestion, the author uses PCA mainly to achieve dimensionality reduction, and the emergence of better clustering characteristics happens to be the characteristics of data presentation. The optimization of LSTM neural network is to achieve better prediction results, improve the overall performance of the recognition model, and prepare for subsequent quantitative analysis.
Point 2: In my experience on using ML for analysis of fluorescence spectra, the simple clustering algorithm will provide the same results.
Response 2: Thanks for the suggestion. Perhaps the ML you use can achieve the recognition of the recognition of the result, the author has introduced in this article, the evaluation index is not a simple classification recognition, but also includes prediction results, accuracy change trend and other indicators to judge the advantages and disadvantages of the model, simply consider the classification results, often neglect other important information.
Point 3: I am not sure is it correct, probably just misleading to say “The original spectral data had 2048 EIGENVALUES”. Anyway, useful part of the spectrum is limited to much less than 2048 points, between 405 nm and less that 700 (650) nm; the spectra should be simply truncated before further analysis.
Response 3:Thanks for the suggestion. The 2048 feature values referred to in this article refer to the 2048 feature variables extracted at specific points throughout the band. The spectra are not truncated in this article due to the integrity of the preserved data. Thank you for your scientific suggestions and corrections.

Reviewer 3 Report
Dear Authors, I have read this text with attention. The use of neural networks to predict mine water accidents is an interesting aspect that the authors of the text have highlighted in the text and demonstrated its potential. The use of laser fluorescence in the study of mine water ejections is an interesting combination of both techniques. These aspects do not raise any objections for me. Alternatively, for applications of this type of technique, I would not limit myself to just sandstone. Mines with limestone and salt mines have a high risk of water ejection.
In general, I agree with the decided theses of the authors, only point out that in this work is left 6 point "patents" but nothing in this paragraph cited. It might also be worth adding more literature items. It is a bit cumbersome to read the cited acronyms, of which there are many in the text. It's worth adding "preliminary research" to the title.
Author Response
Thank you for your instructive comments, the author has made the appropriate changes.
Round 2
Reviewer 1 Report
All my comments have been addressed. The manuscript is suitable for its publication in its current form. I thank The Authors for considering my comments.
Author Response
Thank you for your instructive suggestions, which have gone a long way towards improving this article, and on behalf of the author of this article, I would like to express my sincerest thanks and respect to you.
Reviewer 2 Report
Firstly, in my opinion about the revised manuscript I must repeat my final comment from the first review:
“My final comment is that the strong point of this paper is that is proves that it is possible to obtain correct conclusions from poorly measured results using artificial intelligence.”
Authors should consider to add to the text something like: Our measurements are obtained with no expensive equipment.
In this way, the editor will decide is it manuscript with interesting and sound ML technique applied to poorly measured results worth publishing. In my opinion, Yes.
Author (writing the most of the answers to reviewers using singular) made only technical textual corrections, and provides explanation/answers without changes in text. Some of answers are probably written for editor…. For example:
Point 1: Fluorescence measurements, which are very naive; almost on the verge of being totally incorrect.
Response 1: Thank you for your comments. Regarding point 1, the authors believe that the identification of water sources by laser-induced fluorescence technology is supported by theory and is not a wrong method or viewpoint. Laser-induced fluorescence technology has been used earlier in the academic community to identify edible oil adulteration and combustion diagnosis. Therefore, the use of laser-induced fluorescence technology for water source identification in mine bursts has scientific basis and theoretical support.
Reviewer’s comment:
I didn’t say that using laser-induced fluorescence technology is a wrong method or viewpoint, of course not. I DID SAY that author’s measurements are naïve. Spectra with so much uncorrected background are rather naïve. Moreover using the whole spectra, when there are no indication of upconversion or two photon excitation, is also very naïve, keeping in mind that the optical response to 405 nm excitation is ONLY above the 405 nm…… as confirmed by author:
Point 3: I am not sure is it correct, probably just misleading to say “The original spectral data had 2048 EIGENVALUES”. Anyway, useful part of the spectrum is limited to much less than 2048 points, between 405 nm and less that 700 (650) nm; the spectra should be simply truncated before further analysis.
Response 3:Thanks for the suggestion. The 2048 feature values referred to in this article refer to the 2048 feature variables extracted at specific points throughout the band. The spectra are not truncated in this article due to the integrity of the preserved data. Thank you for your scientific suggestions and corrections.
Looking at Fig 1 it could be seen that measurements of intensity of several hundreds of counts are raised on plateau of at least 400 counts.
Reviewer comment:
There is one point not corrected in text, only mentioned in answers, in regard the fluorescence probe used in the experiment, it turned out that it is not Guangzhou, rather Hangzhou.
Point 6: I could not find data for Guangzhou SPL Photonics FPB-405-V3 immersion probe. I suppose the probe, connected to the Ocean Optics spectrometer, was immersed in water sample,? So the water sample was not placed in cuvette, actually the probe was immersed in the glass bottle? Attenuating the laser excitation by scattering?
Response 6: Thank you for your comments. The fluorescence probe used in the experiment is Hangzhou SPL Photonics FPB-405-V3, not Guangzhou. Experimental methods have mentioned and introduced in the article.
Another reviewer’s comment in regard the ML, for future work:
Now, after I understood that the whole spectra (The spectra are not truncated in this article due to the integrity of the preserved data …. however there is only noisy trash bellow 405 nm and higher than 700 nm; and noised useful signals in range about 405 nm – 700 nm) were used for ML analysis, I must point out that (I had the similar problem in using the ML on fluorescence spectra) it is possible that your ML primarily discerns between different levels of noises of different measurements, not on their different spectral shapes.

Author Response
Thank you for your instructive criticism and suggestions on this article, the author has corrected them accordingly. Regarding your point: the strong point of the paper is to demonstrate that the use of artificial intelligence can lead to correct conclusions from poorly measured results. To this, the authors of this paper concur. The spectrometer used in this paper is not a very expensive piece of equipment, which the authors have added to the article. And this paper is based on the use of this equipment to ensure that the data is true and valid that the experimental methods are reasonable and formal and that the results achieved remain true and complete.
The authors would also like to suggest that the other suggestions you have made are worth considering and exploring. Again, your suggestions are deeply appreciated.